# Effect of High-Density Polyethylene Microplastics on the Survival and Development of Eastern Oyster (*Crassostrea virginica*) Larvae

**DOI:** 10.3390/ijerph20126142

**Published:** 2023-06-16

**Authors:** Sulakshana Bhatt, Chunlei Fan, Ming Liu, Brittany Wolfe-Bryant

**Affiliations:** 1Department of Biology, Morgan State University, Baltimore, MD 21251, USA; 2Patuxent Aquatic and Environmental Research Laboratory (PEARL), Morgan State University, Saint Leonard, MD 20685, USA

**Keywords:** polyethylene microspheres, bivalve, larva, development, survival, ecology, estuary

## Abstract

Microplastic (MP) pollution is a growing global concern—especially in estuarine areas that serve as natural habitats and nurseries for many marine organisms. One such marine organism is the Eastern oyster (*Crassostrea virginica*), which is a reef-forming keystone species in the Chesapeake Bay, the largest estuary in the United States. To understand the potential impacts of MP pollution on the estuary ecosystem, the effects of high-density polyethylene (HDPE) MPs on Eastern oyster larval survival and development were investigated. Three cohorts of larvae were exposed to HDPE MPs with a size of 10–90 µm at a 10 mg/L concentration, after 7 to 11 days of fertilization. After exposure, the number and size of oyster larvae were measured twice a week for approximately 2 weeks until larval settlement. The experiment found that there were no significant differences in the rate of survival between the control and MP-addition treatments. However, we noticed that larval development was significantly delayed with the MP treatment. The percentage of larvae that were ready to settle was 64% with the control treatment compared to 43.5% with the MP treatment. This delay in growth resulted in a delayed larval settlement, which could adversely affect the survival of the Eastern oyster due to the increased risk of predation. The current study demonstrates that MPs could be a risk to the ecology of estuaries, and plastic pollution management is needed for the preservation of these estuaries.

## 1. Introduction

Plastics are synthetic or semisynthetic organic polymers of varied chemical compositions [1]. It has been estimated that each year, over 9.5 million tons of plastics are dumped into the ocean [2]. Based on their size, plastics are divided into Mega plastics (>1 m), Macro-plastics (25–1000 mm), Mesoplastics (5–25 mm) [3], Microplastics (<5 mm), and Nano plastics (<1 µm) [4]. Microplastics (MPs) that are manufactured as microbeads for use in the cosmetic industry are termed primary MPs [5]. Secondary MPs are generated by the fragmentation of larger plastic items due to wave action and UV radiation [6,7,8,9]. Usually, in a marine environment, low-density plastic fragments float on the surface while the high-density ones sink to the bottom—although some low-density plastic particles can also sink to the bottom sediment due to degradation, aggregation, and biofouling [10]. 

Due to the improper management of plastic waste, wastewater discharge containing MPs that evade sewage treatment, and the use of plastic gear in aquatic farming [11], contribute to an increase in MP pollution in the marine environment. Additionally, it has been suggested that rivers transport 88–95% of the MP load to ocean estuaries [12]. Plastics tend to accumulate in the marine environment due to their abundance and persistence [3]. A study conducted on marine surface water particles has shown that a concentration of synthetic and non-synthetic MPs of 11.8 ± 0.6 microparticles/L is present worldwide [13]; MP density in China coastal waters was found to be 0.68 to 6.44 particles/L [14], with concentrations reaching more than 100 g/L reported on Canary Island beaches [15].

Estuaries are highly productive ecosystems where MPs have become available to a wide range of marine organisms [16]. Some MPs, due to their size and resemblance to phytoplankton, are often mistaken for food by organisms [17]. Ingested MPs can also be transferred via predation along the food chain, resulting in bioaccumulation at higher trophic levels. Once ingested, MPs can cause obstruction, abrasion, and physiological alterations that can directly impact the survival of aquatic organisms [2].

Eastern oysters are a reef-forming keystone species found on Chesapeake Bay [18], which is important ecologically as well as economically [19]. Eastern oyster reefs provide many ecological services such as a habitat for many marine organisms; improved water quality through filtration; and shoreline stabilization [19]. Eastern oysters, being filter feeders, have a large potential of ingesting [20] and accumulating MPs in their tissue [2]. Laboratory studies conducted on mussels have reported the translocation of MPs to the circulatory system [21] and accumulation in digestive tissues [22]. A study conducted on edible oysters reported the translocation and accumulation of MPs in the gills, muscles, and digestive glands [23]. Adult Eastern oysters are sedentary and grow attached to the seabed near the shoreline in intertidal regions, where they are exposed to MPs that are sinking or caught in the biofilm [24]. A few laboratory studies have been conducted on the impacts of MPs on Pacific oyster (*Crassostrea gigas*) larvae: A previous study found that Pacific oyster larvae can ingest polystyrene particles of less than 20 µm without any significant impact on their development and feeding capacity [25]; A research conducted on reproductively active Pacific oyster adults and their offspring and reported a decrease in the size and growth of larvae due to polystyrene exposure and a significant decrease in oocyte number and oocyte quality [26]. 

So far, most of the MP exposure studies have been conducted on Pacific oysters and their larvae, and so there is a dearth of data on various MP exposures and their physiological effects on Eastern oysters and their larvae. Therefore, the current study specifically aims to understand the effect of High-Density Polyethylene (HDPE) MPs on the survival and development of Eastern oyster larvae. This ecological study will provide an understanding of the impact of MPs on the population demography of Eastern oysters and will help decision makers to come up with strategies for the restoration of the oyster reef at Chesapeake Bay. This study will also contribute to the optimization of techniques for oyster Hatchery and shellfish culture and the improvement of seafood products, along with the reduction of the use of plastics.

The Eastern oyster life cycle consists of two phases: the fertilization to the larval stages are free-swimming stages, while after larval settlement, the individual becomes benthic [27]. The larval stages are only protected by a soft shell, while adult oysters develop a hard shell for protection. Previous studies have shown that the early stages of development are very critical for the proper development [28] of larvae; therefore, the larval stages of the Eastern oyster life cycle were chosen for the current survival and development study. Ward et al. concluded that bivalves are a poor bioindicator for the quantification of MP pollution in an estuarine environment [29] because they reject some MPs as pseudo-feces, indicating the organisms ability to distinguish between food and non-food items; yet another study has also found MPs in the feces and soft body tissue of Eastern oysters [20], indicating the uptake and retention of plastics by oysters. Eastern oysters are mostly consumed as food—approximately 130,000 tons/year are produced and marketed [2]—and therefore they can also be used as a model for human health risk assessment due to seafood consumption. Moreover, Eastern oysters are reef-forming keystone species; therefore, their population dynamics are a good indicator of coastal health and a coast’s ecological stability. 

The current study chose to investigate ecological parameters such as the survival of Eastern oyster larvae because the survival of the larvae enhances their chances of successful settlement and contribution towards reproduction to increase the population size. The second reason for choosing the survival of the larvae for this study is because previously, a study conducted previously observed an abnormal swimming behavior of the Pacific oyster larvae caused by HDPE-MPs exposure and suggested that this could lead to an adverse effect on the survival of the larvae [30]; therefore, here the current study investigated the effect of MPs on the survival of Eastern oyster larvae. Additionally, MPs are of varied chemical compositions. Polyethylene was chosen for this study because polyethylene is one of the most abundant types of MPs in wastewater [31] that arises from urban waste and civil engineering projects [23]. It is estimated that polyethylene alone contributes to almost two-fifths of the total plastic waste in the marine environment [32]. Furthermore, HDPE microspheres were chosen over LDPE microspheres because LDPE mostly floats at the water surface and does not sink to the bottom of the water column, while HDPE microspheres can be easily percolated to the entire water column, including sinking to the bottom. Additionally, a study conducted previously has shown that HDPE-MPs can accumulate in the digestive gland, gills, gonads, and muscle of mussels [33]—indicating that they have the capacity to be ingested by aquatic organisms and show subsequent bioaccumulation.

## 2. Materials and Methods

### 2.1. Experimental Materials

HDPE-MPs of white color, with a diameter of 10–90 µm [2,34,35] (Cospheric), were used in the survival and growth experiments. The density of the polyethylene microspheres was 1.35 g/cc to ensure that the microspheres not only floated on the water’s surface but were also submerged in the water. The aeration system was used to continuously percolate the MPs’ microspheres throughout the water column. A previous study conducted on mussels has shown that MP beads are ingested easily and can be transferred and accumulated in the digestive tract due to their small size [14]. Another study on the ingestion of regular-shaped microbeads versus irregular-shaped MPs reported a significantly low amount of ingestion of irregular-shaped MPs compared to regular-shaped MPs at a concentration of 10 g/L [36]. A HDPE-MP concentration of 10 mg/L [2,11,30,37,38] was used for the MP exposure experiments for the current study. Additionally, a study conducted on MP ingestion by bivalve veliger larvae reported the uptake of MPs at 10 mg/L [39]. 

### 2.2. Hatchery Environment

The HDPE-MP exposure experiments were conducted in controlled Hatchery conditions at the Morgan State University Patuxent Environmental and Aquatic Research Laboratory (PEARL). The seawater was pumped from the Patuxent River and was filtered using a sand filtration system down to 2 µm, followed by cartridge filtration to 0.5 µm filters to minimize the effect of predation and disease on the growing larvae. Salt was added to the filtered seawater to increase the salinity of the seawater to 14 ppt. The water’s pH was monitored every 2 to 3 days and was kept around pH 8. The oxygen levels were monitored regularly, and the water was aerated continuously. The larvae were fed with laboratory-cultivated algae—mainly *Isochrysis galbana, Chaetoceros muelleri*, and *Tetraselmis chui*.

### 2.3. Spawning

The Eastern oysters were collected from the Patuxent River. In the hatchery, the microscopic gametes of male and female Eastern oysters were harvested by strip spawning for external fertilization in the filtered seawater. Usually, fertilized gametes grow into free-swimming larvae known as Trochophores, with cilia and a small shell, in approximately 6 h. Trochophores develop a hinged side and a velum within 12 to 24 h and are then known as D-shaped larvae or veligers. The veligers grow for 2 to 3 weeks—swimming freely—and reach a size of between 200 µm and 300 µm, developing an eyespot and a foot and are then known as a pediveliger; at this stage, the larvae are ready to settle to a substratum, and once they attach themselves to a surface, they are then referred to as Spats.

### 2.4. Experimental Design 

Four large buckets (two for the control treatment, and two for the MP exposure treatment) with a capacity of 25 L each were used to hold the larvae. All the buckets had approximately the same number of larvae and were maintained at a specific density. At day 0, the stocking density was 10 larvae/mL, from day 2 onwards 5 larvae/mL, from day 7 onwards 4 larvae/mL, from day 12 onwards 3 larvae/mL, and from day 14 onwards 2.5 larvae/mL. All buckets were supplied with continuous airflow. The temperature, salinity, pH, and dissolved oxygen concentration were monitored for the entire duration of the experiment. After every 2–5 days, the number of surviving larvae was counted by volumetric methods using a Sedgewick rafter cell 550 (Graticules optics) and Standard lab light microscope (Leitz Laborlux D). At each count, the size of 10 larvae from each group was also measured by treating the larvae with a drop of isopropyl alcohol (20 mL 70% isopropyl alcohol dissolved in 30 mL of DI water) to slow the speed of the actively swimming larvae, and by using the eyepiece reticle attached to a light microscope. 

### 2.5. Analytical Method 

The first and second cohort of larvae were produced at the MSU PEARL hatchery. The second cohort of larvae was produced because of the mortality issues with the first cohort of larvae. For each cohort, the larvae were counted using volumetric methods and then both cohorts of larvae were randomly separated into four groups. The experiment was conducted in duplicate. The two groups were exposed to HDPE-MP beads with sizes ranging from 10–30 µm, and the other two groups served as the control for the experiment. The survival rate was calculated by taking the total number of larvae and dividing it by the previous total number of larvae to calculate the survival from one water change to the next. However, the larvae in Cohort 1 and Cohort 2, all perished at the veliger stage of development.

### 2.6. Settlement Study 

To further study the effect of HDPE-MPs on this stage transition of oyster larvae, a cohort of Eastern oyster larvae was purchased from Piney Point Aquaculture Center (MD). These larvae (Cohort 3) were from hatchery-grown diploid oysters and were an average of 153 µm long on day 8 after fertilization. The experimental procedure is the same as the first two cohort experiments. The larvae were divided into four groups. The experiment was conducted in duplicate. Two groups were exposed to HDPE-MPs of sizes ranging 10–90 µm, with a concentration of 10 mg/L; then, the larval survival rates and size were measured three times a week. Since some of these larvae reached a size greater than 298 µm and developed eyespots and feet on day 16 post-fertilization, the percentage of larvae that were ready to settle was also measured using the 211-micrometer sieves at the later stage of this experiment. 

All the MPs left in the water were collected on 10 µm filter sleeves at every water change to prevent the MPs from releasing into the environment.

### 2.7. Statistical Analysis

All experiments with the 3 cohorts of Eastern oyster larvae were conducted in duplicate; the data for the surviving larvae were divided by the total larvae at every count to obtain the survival rate. Statistical analyses were conducted with R using an α value of 0.05. A significant difference was accepted when *p* < 0.05. A paired t-test was used to compare the percentage of larvae that were ready to settle from the control and experimental groups.

## 3. Results

The larval survival rates of the three cohort larval experiments after the MP exposure is shown in Figure 1 (Appendix A). Overall, the larval survival rates exhibited a large variation during the experimental period, and there was no significant difference in the overall survival rate between the control and the MP-exposure treatments (*p* > 0.05). The survival rate of Cohort 1 larvae on day 3 of HDPE-MP exposure was significantly higher with the control treatment (~57%) than with the MP treatment (~36%). However, on day 5 of HDPE-MP exposure, the larval survival rate with the MP treatment was higher than with the control treatment. A similar pattern was also observed for the Cohort 2 experiment; the larval survival rates of Cohort 2 were generally lower compared to the Cohort 1 and Cohort 3 experiments. In Cohort 2, on day 4 and day 9 of MP exposure, the larval survival rates in the control group were slightly higher than with the MP treatment. All the larvae in the MP treatment groups perished on day 11 of exposure, while the survival rates for the control groups were still at 15%.

The larvae in Cohort 1 and 2 experiments all perished around 10 days after MP exposure. Thus, another cohort of larvae was purchased to continue the MP exposure experiment. In this Cohort 3 experiment, the larval survival rates were generally higher compared to Cohort 1 and Cohort 2, as these larvae were bigger in size (Figure 2); there was no significant difference between the control and MP treatments on day 4, 6, and 7 after MP exposure. The only difference was observed on day 10, when the larval survival rates with the control treatment were higher than with the MP treatment. 

The average larval size increase of the three cohorts during the exposure study is shown in Figure 2. This figure shows the larval growth and development pattern over the experiment. The larvae in Cohorts 1 and 2 were produced from a hatchery at MSU PEARL. At the beginning of the experiment, the larvae were 7 to 11 days post-fertilization, with an average larval size of 82 µm in Cohort 1 and 78 µm in Cohort 2, respectively. For the Cohort 1 larvae, a significant difference in the average larval size between the control and MP treatments was observed on days 3, 5, and 7 after MP exposure (*p* < 0.05). However, the larval size did not increase for both the control and MPs treatment over most of the experimental period; the only increase was observed for the control treatment at day 10 after MP exposure, when the larval size reached 112 µm. The growth and development pattern for Cohort 2 was similar to the Cohort 1 experiment; there was no significant increase in larval size during the experiment. Overall, the larval size results in Cohorts 1 and 2 suggested that the larvae did not grow or develop in these two experiments, and some other environmental factors could hinder larval development in these two experiments. 

Since the first two cohorts of larvae all perished before reaching the pediveliger stage, the Cohort 3 larva were purchased from another hatchery facility to continue the MP exposure experiment. At the beginning of the experiment, the larvae were 8 days post-fertilization with an average length of 153 µm, which was significantly larger than the other two cohorts. Overall, the average larval size increased significantly over the experimental period, with the control treatment reaching ~325 µm and the MPs treatment reaching ~300 µm, respectively. Furthermore, toward the later part of the experiment on days 6, 7, and 10, the larval size in the control group was significantly higher than with the MP treatment (*p* < 0.05); this clearly indicates that larvae were growing faster in the control groups as compared to the MP groups. 

Starting day 6 of the Cohort 3 experiment, the larval size reached above 200 µm, and some larvae developed an eyespot and a foot and were ready to settle to a substratum. This portion of larvae was collected using the 211-micrometer sieves and was treated as larvae that were ready to settle. Figure 3 combines all the data from days 6, 7, and 10 after MP exposure. The paired *t*-test showed a significant difference in the percentage of larvae ready to settle between the control and HDPE-MP treatments (*p* < 0.05). In agreement with the average larval size results (Figure 2), the percentage of larvae ready to settle with the control treatment was significantly higher than with the MP treatment. This again confirms that the larvae in the control group were growing faster and more were ready to move from the larval stage to a spat stage as compared to the MP treatment group.

## 4. Discussion

All three cohorts of experiments lasted approximately two weeks after MP exposure. Compared to previous studies that were mostly conducted in a relatively short time period (e.g., 24 h) [11,30], this study covered most of the free-swimming stage of Eastern oyster larvae, and could be a better representation of the impacts of MPs on larval survival and development. Even with a large variation over the experimental period, the quantitative data in the current study clearly indicates that although the rate of survival was not significantly different between the two groups, the growth of the larva was significantly affected by the HDPE-MP exposure. This retardation in growth could be very costly from the population’s perspective, because it increases the chances of larval predation and thus decreases the reproductive pool of the population. The Eastern oyster population has declined over the past 130 years [27] and the findings of the current study suggest that the growth retardation of the larva due to MP exposure could be one of the many reasons for this decline in recent decades.

A previous study conducted on Pacific oysters reported that MP (10 mg/L)-exposed larvae have a low rate of settlement, with significantly lower growth for the first 28 days of development [2]; this current study is in agreement with the observation that larval growth slows down significantly due to HDPE-MP exposure. Research conducted previously has shown that nano-size plastics can get stuck on locomotor eyelashes [30]—this accumulation around the velum might interfere with the feeding of the larvae. In the current study, it was observed—under the microscope—that there was an agglomeration of MPs around a few larvae shells that might impair proper feeding, causing a nutritional deficiency and resulting in slow body growth.

In the current study, it was observed that larvae frequently collided with large MP microspheres during their swimming activities—this might cause them to spend extra energy getting around the obstruction caused by the MPs and thus have less energy available for their growth and development. Moreover, studies conducted on the effect of MPs on aquatic organisms have reported decreased energy reserves in some bivalves and marine worms [40,41], as well as a decrease in lipid content in bivalves due to MP exposure [40,42]. Additionally, a study conducted previously concluded that the uptake and elimination of MPs is an energetically costly process that results in smaller organism sizes [43]. Altogether, MP exposure results in higher maintenance costs and less energy being available for other physiological functions.

A study on Pacific oyster larva reported the propensity of MP consumption and concluded that larval age and the MPs’ size affected the amount of plastic consumption [25]. Additionally, one study conducted on post-oral processing in bivalve larvae has shown that the larvae can ingest and retain microbeads in their gut [44], indicating the bioaccumulation of microbeads in the gut lumen. A study on Pacific oyster larva reported that MPs tend to accumulate in the larval digestive tract in a dose-response manner [11]. On average, the clearance of MPs from the gut of larvae are reported for 24 h after exposure; however, studies have shown that it takes 8 days for 100% gut clearance [45], indicating the resistance to elimination and a tendency towards accumulation. This might cause intestinal blockage and may damage the wall of the digestive tract. This in turn will result in the poor growth of the organisms.

The current study did not find any statistically significant differences in the survival rate of larvae, but that could be due to the larger particles of HDPE-MPs used in this experiment, the difference in the densities of the particles, or simply because the experiment was conducted on a different species of oyster. Furthermore, the larvae were produced for survival studies by stripped spawning, which sometimes results in poorly developed or malformed eggs [46], which might affect the results. 

The first two cohorts of larvae perished after about 10 days of MP exposure, and never reached the pediveliger stage at which they would be ready to settle. This unsuccessful growth and development could be the result of the presence of some microscopic organisms that prey on early-stage oyster larvae in this experiment. To minimize the effect of disease and predation on Eastern oyster larvae in control and experimental buckets, filtered sea water was used and the water was changed after every 2 to 4 days. However, the presence of microscopic organisms was frequently noticed—such as species of rotifers and protozoa—when the number of oyster larvae were counted in the experiment. Some previous research has also suggested that these microscopic organisms could influence larval growth and development [47]; this is another confounding variable of the current study and could have contributed to the results that were observed.

A study conducted on pearl oyster larvae has shown that 48 h exposure to MPs significantly increases larval mortality [2]. Similarly, a study conducted on Grunion larvae reported high mortality rates due to MP exposure [48]. Additionally, a study conducted on adult Pacific oysters also reported increased mortality rates due to MP exposure [37]. In the current study, no significant mortality of larvae was observed due to MP exposure. One reason for this might be that most of the studies conducted on these larvae use MPs that are less than 10 µm in size, and only a few studies are conducted above 10 µm, up to a maximum of 25 µm [11,30].

Future study goals include investigating the impact of MPs of varied chemical compositions on Eastern oysters. MPs will be extracted from Eastern oysters and AIM 9000 optical FTIR (Shimadzu) will be used to calculate the abundance of ingested MPs and relate this to any observed physiological effects.

Wherever possible, we minimize the use of plastic materials for this study, but a lot of hatchery equipment and water pipes are made of plastics and their use cannot be avoided. One serious limitation of this study was the use of pristine MPs instead of environmentally aged particles.

## 5. Conclusions

The present ecological study evaluates the Chesapeake Bay ecological risk assessment due to MP pollution by studying the impact of MPs on the growth and survival of Eastern oysters. Here, understanding the effects of MP pollution on Eastern oysters will help to mitigate strategies for reef restoration and increase the amount of seafood available for human consumption. In summary, this study did not find any statistically significant difference in the rate of the survival of Eastern oyster larvae when they were exposed to HDPE-MPs (at a concentration of 10 mg/L) for an exposure time of 3 to 13 days. However, HDPE-MP (at a concentration of 10 mg/L) exposure for 13 days affected the growth of the Eastern oyster larvae. For larvae that were exposed to HDPE-MPs, it took a longer time to reach the settlement stage. Although we did not find any significant difference in survival rate in the laboratory study, the larvae’s delayed growth in the natural environment increases the chance of larval predation, and thus reduces their survival. The more larvae that fall to predation, the greater the reduction in population size, resulting in demographic transitions.

## Figures and Tables

**Figure 1 ijerph-20-06142-f001:**
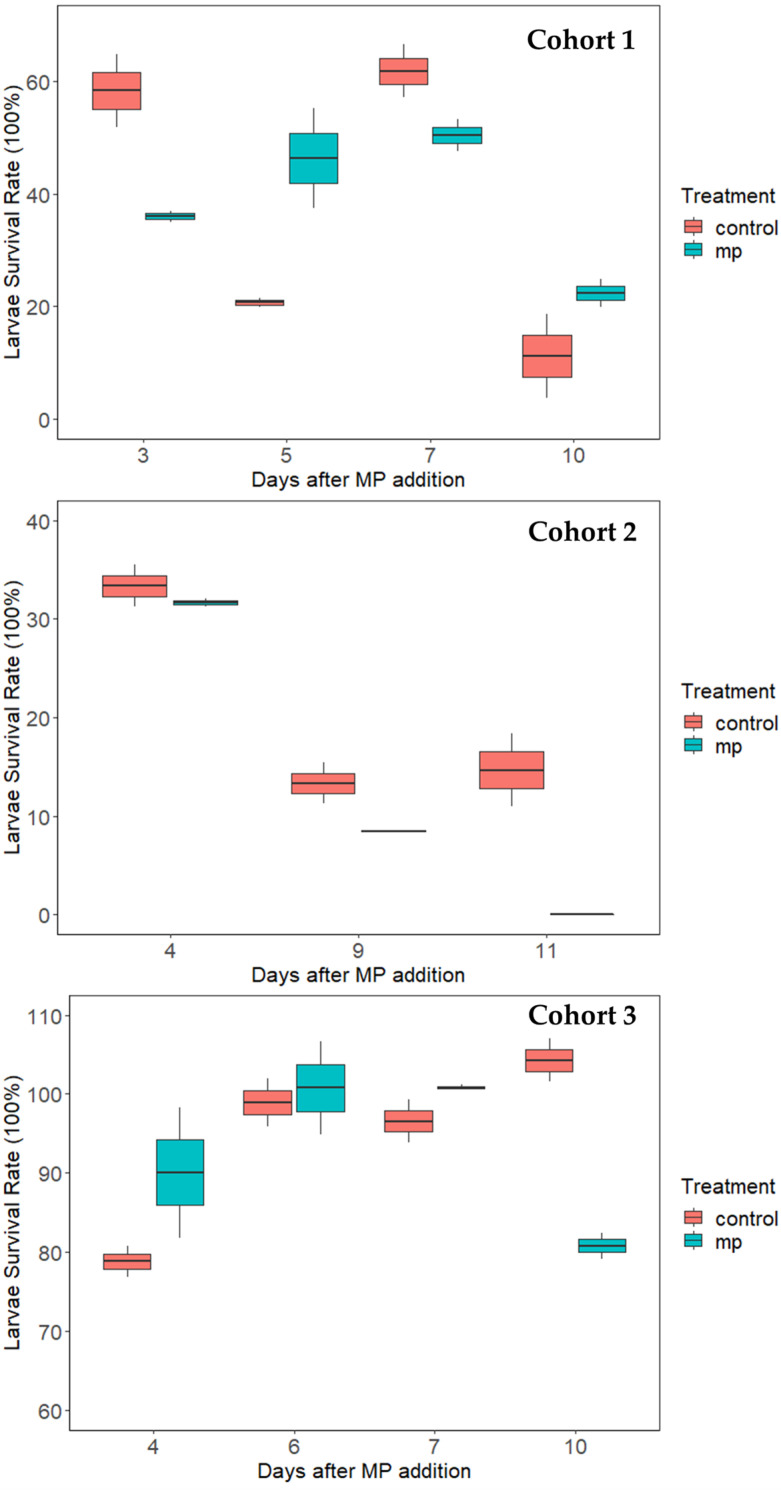
Boxplot of the survival rates of *C. virginica* after high-density polyethylene microplastic exposure in the three cohorts of larval experiments. For *n* = 2, the horizontal line inside the box represents the median value.

**Figure 2 ijerph-20-06142-f002:**
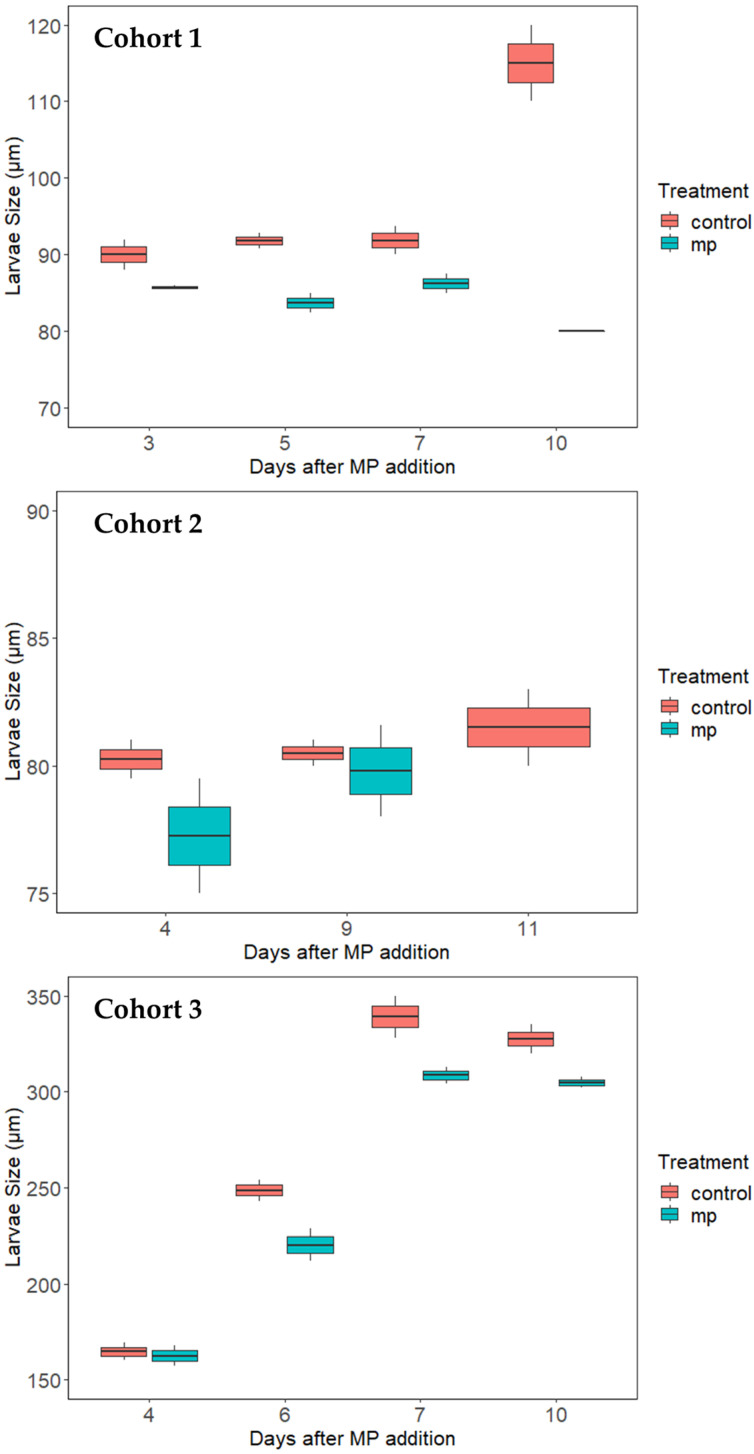
Boxplot of the size of larvae of *C. virginica* after high-density polyethylene microplastic exposure in the three cohorts of larval experiments. The horizontal line in the box represents the median value (*n* = 2).

**Figure 3 ijerph-20-06142-f003:**
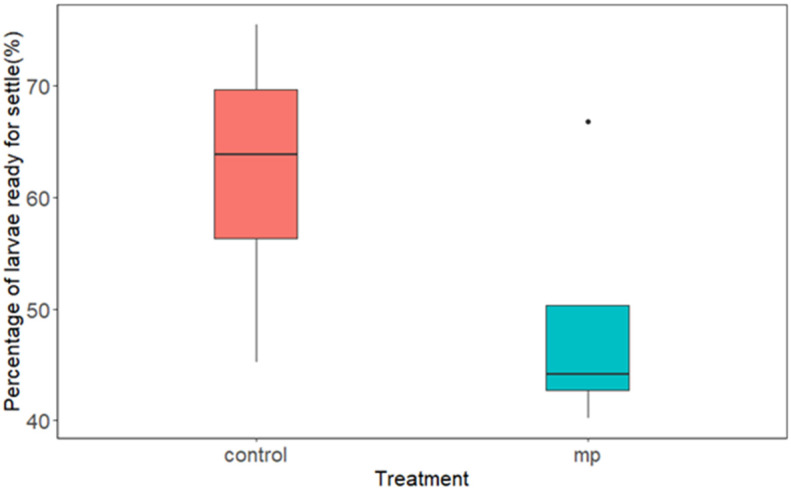
The percentage of *C. virginica* larvae that were ready for settlement in the Cohort 3 experiment.

## Data Availability

The data presented in this study are stored at Patuxent Environmental and Aquatic Research Laboratory and can be obtained upon request.

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
