# Peer review of "Effect of High-Density Polyethylene Microplastics on the Survival and Development of Eastern Oyster (Crassostrea virginica) Larvae"

_ijerph, 2023, doi:10.3390/ijerph20126142_

Round 1

Reviewer 1 Report

In this manuscript, several concerns affect the validity of the scientific conclusion of the study as follows:

1. The authors assessed the effect of polyethylene microplastics on Eastern Oyster (Crassostrea virginica) Larvae number and size. The estimated parameters are too preliminary. The assessment of the changes in the physiological indicators and determination of the microplastic content in the tissues are needed to give a valid scientific conclusion.

2. The design of the experiment is unclear and confusing. Why two control groups were used? Why the concentration of 10 mg/L was used? It is not enough to just give a reference. The author should justify in detail the tested concentration. 

3. The presentation of figures is unclear, low quality, and not reflecting the statistical analysis. Also, the figure legend is very concise and nor reflect the number of replicates or how the data were presented.

4. The abstract lacks quantitative information about the research methods, and the presentation is confusing. There is a paucity of detail about the experimental protocols, particularly the diameter and concentration of microplastics.

5. Line 213 and 219: the paragraphs begin with studies. Thus, more than one reference should be present at the end of the paragraph.

6. It is not preferable to begin sentences with abbreviations like MPs in line 45.

7. There is a problem with using abbreviations throughout the manuscript. The full term should be mentioned first with the abbreviation between paresis then the abbreviations should be exclusively used throughout the manuscript. E.g., in line 30, Microplastics (MPs), the full term has been mentioned again in lines 31, 68, 75, 219, 229, 231, …etc. Such errors have been repeated for many abbreviations throughout the manuscript.

8. The writing style should be formal from the third-person perspective. Do not use we or our (E.g. line 13, “, we investigate the effects” should be “the effects were investigated”; line 17, “Our experiment” should be “The current experiment”).

Author Response

  1.  We changed the "physiological indicators" to ecological indicators in the manuscript to avoid confusion. The authors are not sure what type of "physiological indicators" the reviewer would like to have. This study is to assess the impacts of microplastic on the larval survival and development. Given the smaller size of oyster larvae, it is impossible to measure the microplastic content in the oyster tissues. 
  2. There were no two control groups in the experimental design. There were two treatments (e.g. control and MP addition treatments). For the control treatment, we had duplicate samples. to MP addition concentration (10mg/L) was based on the previous similar research, we also added the discussion on this concentration in the revision.
  3. In the revision, we improved the resolution of the figures. The figures are simple boxplots of the survival rate, size, and percentage of larvae ready for settlement in our experiment. Two treatments (control and MP addition) were indicated as a legend. Each treatment had duplicate samples, the boxplot represents the middle 50% of the data.
  4. The abstract has been revised based on the suggestions. Quantitative information regarding the size and concentration has been inserted along with the developmental results.

  5. In lines 213 and 219, the studies are replaced by “The study" or "study"

  6. In Line 45, the sentence with MPs has been corrected by inserting "Some" before MPs.
  7. The abbreviations throughout the manuscript have been revised and corrected. The full term is mentioned only once and the corrections in lines 31, 68, 75, 219, and 231 were made by changing them to the abbreviated form.

  8. The writing style throughout the manuscript has been revised and corrected and all “we” and “our” are replaced as suggested by the Reviewer. The sentences in line 13 and line 17 are rephrased according to the reviewer’s suggestions.

Reviewer 2 Report

The current manuscript entitled “Effect of High-Density Polyethylene Microplastics on Survival and Development of Eastern Oyster (Crassostrea virginica) Larvae” by Bhatt et al. deals with the effects of high-density polyethylene microplastics (HDPE-MPs) on the Eastern oyster larval survival and development. The overall design of the experiment was appropriate to answer the research question, although some methodological improvements could have been made. For example, the authors did not provide any information on the microscopic analysis of the microplastics used in the experiment. It is possible that the dose of the microplastics could have been a factor in the results of the experiment, and this should have been taken into consideration. I suggest major revision and my specific comments are:

1.      Authors should write HDPE-MPs consistently instead of using several versions like HDPE microplastics, MPs, etc.

2.      The abstract lacks major numerical findings of this study and needs to be revised accordingly.

3.      Please avoid all personal words such as we, us, our, etc.

4.      The keywords should be modified to avoid those which already appeared in the title.

5.      Put heading/subheading numbers.

6.      Reference selection is poor, for example, the whole introduction contains several arguments which repeatedly are supported by 1 or 2 references. Also, the style of reference is not as per the journal. Please add valid references to almost all claims authors made in the whole manuscript except their own findings.

7.      The language of the manuscript must be improved, the current version contains immeasurable errors in terms of spacing, punctuation, grammar, and syntax.

8.      The “material and method” section must be categorized into several subheadings, e.g., Experimental Materials, Experimental Procedure, Analytical methods, data analysis and software, etc.

9.      The source and manufacturing information of HDPE and other materials are missing. Why did the authors not conduct an SEM analysis of the HDPE?

10.   The treatment design is confusing, why only one level 10mg/L of MP was tested for 10-30 um?

11.   Add scientific authorities for the scientific names of organisms mentioned.

12.    The data provided is in % and not actual, It would be desirable to provide actual experimental values in tables also and conduct an appropriate test of significance in order to understand what effects MPs brought.

13.   Colorize Fig. 3 for consistency.

14.   Increase the number of references. The same references are repeated in the discussion section which is not advisable. Several claims in the discussion are ambiguous and need to be revised with appropriate citations. I can’t mention them all here.

15.   Conclusions made based on the findings don’t show any novelty. Better to rewrite this section.

Author Response

  1. Authors have corrected the HDPE-microplastics to HDPE-MPs throughout the manuscript, however, there are sentences that talk about MPs in general and not particularly HDPE, those places we have only MPs and not HDPE-MPs.

  2. The survival rates varied significantly over the two-week experiment period, so we only provide the statistical results without the numerical values of the survival rate. However, we revised the abstract and added the numerical values (percentage) of larvae ready to settle.

  3. The writing style throughout the manuscript has been revised and corrected and all “we” and “our” are replaced as suggested by the Reviewer.

  4. The keywords have been modified and new words are added that are not in the title.

  5. The numbers to headings and subheadings have been added.

  6. The reference section has been improved considerably. As per Reviewer’s suggestion, to add valid references to all claims in the introduction section, the arguments in the introduction section are now supported by a reference.

  7. The language and grammatical corrections to the entire manuscript have been made.
  8. In the Material and Method section, new subheadings have been created that are as follows.

    Experimental Materials, Hatchery Environment, Spawning, Experimental Design, Analytical Method, and Settlement Study.

  9. Microplastic microspheres that we used in our experiment were commercially obtained from Cospheric. They were uniform in chemical composition, and the size ranges were clearly specified, so there was no need for formal SEM analysis.

  10. The dose/concentration of the MP could be a factor that influences larval survival and development. But in our experiment, we want to find out whether the MP could impact these larval ecological indicators, and further study of MP concentration could be performed.

  11. The authors were not able to understand fully the comment "Add scientific authorities for the scientific names of organisms mentioned". The binomial nomenclature has been followed to write the scientific names of Eastern and Pacific oysters the first time they appear in the text. The abbreviated scientific name of the Eastern oyster (C.virginica) has been used in the figures and for the rest of the text, a common name is used to refer to the organism.

  12. Our experiments lasted two weeks, and the number and size of live larvae (actual experimental values) were counted twice a week. Then the % value (survival rate) was calculated based on these actual values. The actual values / raw are now provided in the supplementary part.

  13. Fig. 3 has been improved by adding color to the box plots.

  14. The reference section has been improved by adding more references. The discussion section has been improved by rewriting the section according to the Reviewer’s suggestions.

  15. The conclusion section was improved based on Reviewer’s suggestions. The novelty in our study is that our research follows the whole larval stage of the Eastern oyster and tries to assess how the MP can impact its survival and development. Compared to most of the previous studies, which only last a short time period, our study is a better representation of how MP could actually impact the Eastern oyster larval survival and development.

Reviewer 3 Report

The manuscript is quite well written but requires some important (major) revisions before publication. This is especially true of the discussion, which is weak and needs to be improved.

List of comments

Line:

38                     “small MPs” - The definition of MPs has been given previously. The use of the words "small" or "large" MPs is redundant.

50                     Easter oyster”  -  or Eastern oyster?

52-53              Is MPs absorbed in the oyster's digestive system and accumulated in tissues? Or rather accumulating in the digestive system? Please give more attention to this topic in the introduction.

66, 76             “beads”  - or irregular pellets? Only cospheric beads were used in your research. Why was this shape chosen? Can the shape of the polyethylene beads affect the test results? Please expand on this topic in the introduction and discussion. Examples of literature: https://doi.org/10.1016/j.scitotenv.2019.01.281

67                    Survival rate and the size of the oyster larvae are not an eco-physiological parameters (as physiological parameters of oyster larvae have not been studied in yours). In my opinion, the survival rate and the size of oyster larvae are an ecological parameters.

80                     Why was this concentration chosen? (what is the average concentration in estuarine waters?)

102                  “were maintained at reasonable density”  - What was the concentration of larvae? This is key information to which the number of surviving larvae should be related. Please use more precise and scientific language!

116-117          “However, in both experiments, all larvae perished at the veliger stage of development.”  -  “in both experiments” Cohort 1 and 2 or in both groups with MPs treatment? Please explain.

147-150          “On day 4 and day 9 after the MP exposure, the larval survival rates in the control group were slightly higher than in the MP treatment. All the larvae in the MP treatment groups perished on 11 days of exposure while the survival rates for the control groups were still at 15%.” -  Which cohorts? Please be more precise. (I know that the term 11 day was only in 2 cohorts, but it is a mental shortcut that should be avoided in the description of the results.)

202-219          This is not a discussion but an introduction.

219-220          The cited studies ([12] Craig et al. 2022) determined the egestion efficiency of MPs in faeces and pseudofaeces. MPs may retention in the digestive system, but why the conclusion about the accumulation of MPs in soft tissues? Can MPs be absorbed in the digestive system of oysters and accumulated in tissues?

221-232          This is not a discussion but an introduction.

236-239          The effect of nanoplastics on organisms is a separate scientific issue in relation to MPs. The discussion should be an attempt to explain the mechanism of action of MPs on the abnormal development and mortality of oysters.

240-247          This is not a discussion but an introduction, possibly an introduction/justification to the methodology.

247-250          This is a good snippet of discussion confirming the accumulation and explaining the reasons for the negative impact of MPs on oyster larvae. Please expand this thread (look for similar research results).

251                  Survival rate and the size of the oyster larvae are not an eco-physiological parameters (as physiological parameters of oyster larvae have not been studied in yours). In my opinion, the survival rate and the size of oyster larvae are an ecological parameters.

251-258          This is not a discussion but an introduction, possibly an introduction/justification to the methodology.

264-267         This is not a discussion but an conclusion.

278-287          This is a good snippet of discussion and conclusion.  Please expand this thread (look for similar research results).

Best regards

Author Response

  1. Line 38 “small MPs” have been corrected by removing the small before the MPs.
  2. Line:50   “Easter oyster has been corrected to Eastern oyster.
  3. Lines: 52-53 This initial line is now supported by information regarding MPs accumulation and translocation in mussels and edible oysters in the introduction section.
  4. In Lines: 66, and76 the word beads has been replaced by microspheres. This shape was chosen because it was commercially available and the literature shows that aquatic organisms easily ingest this shape. Also, MPs beads are abundantly used in the cosmetic industry. A sentence and supporting citation have been added in the Material and Method section regarding using beads/microspheres in the current study. The shape, surface, and chemical compositions can have an effect on the uptake of MPs and will be part of future studies. In the current study, the surface of the microspheres was smooth and the shape was spherical observed under the light microscope. A justification has been added to the Material and Method section for the use of microspheres that are sometimes referred to as beads and regularly shaped microplastics in the literature.

  5. In Line 67: the "eco-physiological" has been changed to "ecological" study.
  6. In Line 80: the concentration of 10 mg/L was based on the previous research, the authors also added a discussion on this concentration in the revision.
  7. In Line 102: the density of the larva in the control and experimental buckets is now fully explained in the Material and Method section.
  8. In Lines 116-117: “However, in both experiments, all larvae perished at the veliger stage of development.” To improve the clarity of this sentence Cohort 1 and Cohort 2 have been added to the sentence.
  9. To improve the clarity of the sentence in Lines 147-150 Cohort 2 has been added to the sentence.
  10. Lines 202-219 have been removed from the discussion, they are now part of the Introduction.
  11. Lines 219-220 are now part of the Introduction and this discussion thread has been modified for more clarity.
  12. Lines 221-232 are moved from the Discussion section and are now the part of Introduction.
  13. In Lines 236-239, the authors feel that the observation made by the use of Nanoparticles in a previous study is necessary to discuss because similar observations were made in the current study under the light microscope. This thread of discussion has been expanded to include our observations.
  14. Lines: 240-247 have been moved from the discussion section to the introduction section.
  15. Lines: 247-250 have been moved from the discussion section to the Introduction section. The discussion section is rewritten explaining the negative impact of MPs on the growth of oyster larvae. 
  16. Line 251 the "ecophysiological" is changed to ecological.
  17. Lines 251-258 have been moved to the Introduction section.
  18. Lines 264-267  have been moved to the conclusion section as per the suggestion.
  19. Lines 278-287 Two more references have been added to discuss the survival rate due to MPs exposure.

Round 2

Reviewer 1 Report

The authors have made some revisions to the manuscript. Still, two concerns exist:

1.            Despite the improvement in figure quality, the legends need to be informative. In all figure legends: clarify if the data has been presented as means±SE or SD and the number of replicates n=? Also, the full term of all abbreviations used should be clarified.

2.      Throughout the manuscript, either use MP or MPs.

Author Response

  1. To enhance the clarity of the figures, the following statement has been added to Figure 1 and Figure 2.

The box represents the middle 50% of the data (interquartile range), with the median value (n=2) as a horizontal line.

In the description of Figures 1 and 2, the abbreviation HDPE-MPs has been replaced by high-density polyethylene microplastics.

2. MPs replace all MP throughout the manuscript.

Reviewer 2 Report

Authors have revised the manuscript as per my comments. I have no further comment. The manuscript can be accepted in current form.

Author Response

The reviewer does not have any further comments after the revision.